# Poorer Survival in Patients with Cecum Cancer Compared with Sigmoid Colon Cancer

**DOI:** 10.3390/medicina59010045

**Published:** 2022-12-27

**Authors:** Shibo Song, Jiefu Wang, Heng Zhou, Wenpeng Wang, Dalu Kong

**Affiliations:** 1Department of Endoscopy, National Cancer Center/National Clinical Research Center for Cancer/Cancer Hospital, Chinese Academy of Medical Sciences and Peking Union Medical College, Beijing 100021, China; 2Department of Colorectal Oncology, Tianjin Medical University Cancer Institute and Hospital, National Clinical Research Center for Cancer, Tianjin’s Clinical Research Center for Cancer, Key Laboratory of Cancer Prevention and Therapy, Tianjin 300060, China; 3Department of Oncology Surgery, People’s Hospital of QingXian, Cangzhou 062655, China

**Keywords:** prognosis, colon cancer, cecum cancer, sigmoid colon cancer, compare

## Abstract

*Background and Objectives:* An increasing number of studies have shown the influence of primary tumor location of colon cancer on prognosis, but the prognostic difference between colon cancers at different locations remains controversial. After comparing the prognostic differences between left-sided and right-sided colon cancer, the study subdivided left-sided and right-sided colon cancer into three parts, respectively, and explored which parts had the most significant prognostic differences, with the aim to further analyze the prognostic significance of primary locations of colon cancer. *Materials and Methods:* Clinicopathological data of patients with colon cancer who underwent radical surgery from the Surveillance, Epidemiology, and End Results Program database were analyzed. The data was divided into two groups (2004–2009 and 2010–2015) based on time intervals. Two tumor locations with the most significant survival difference were explored by using Cox regression analyses. The prognostic difference of the two locations was further verified in survival analyses after propensity score matching. *Results:* Patients with right-sided colon cancer had worse cancer-specific and overall survival compared to left-sided colon cancer. Survival difference between cecum cancer and sigmoid colon cancer was found to be the most significant among six tumor locations in both 2004–2009 and 2010–2015 time periods. After propensity score matching, multivariate analyses showed that cecum cancer was an independent unfavorable factor for cancer specific survival (HR [95% CI]: 1.11 [1.04–1.17], *p* = 0.001 for 2004–2009; HR [95% CI]: 1.23 [1.13–1.33], *p* < 0.001 for 2010–2015) and overall survival (HR [95% CI]: 1.09 [1.04–1.14], *p* < 0.001 for 2004–2009; HR [95% CI]: 1.09 [1.04–1.14], *p* < 0.001 for 2010–2015) compared to sigmoid colon cancer. *Conclusions:* The study indicates the prognosis of cecum cancer is worse than that of sigmoid colon. The current dichotomy model (right-sided vs. left-sided colon) may be inappropriate for the study of colon cancer.

## 1. Introduction

In the era of personalized and precise treatment, due to differences in anatomy and histology between right-sided and left-sided colon, increasing studies have begun to explore whether different primary tumor locations for colon cancer have a significant impact on prognosis [1,2,3,4,5]. Most previous studies have shown that patients with right-sided colon cancer (RCC) have a poorer prognosis compared with left-sided colon cancer (LCC) [1,2,3,4,6,7,8,9], but others are inconsistent [10,11,12,13]. Warschkow et al. found a better survival outcome in RCC relative to LCC among patients with stage I–III colon cancer using the propensity score matched (PSM) method [11]. Therefore, there is still a controversy in survival differences between RCC and LCC. 

Up to now, the definition of RCC has been divided because some studies classified the transverse colon as right-sided colon, but others excluded it directly [1,8,9,10,11,13,14]. In addition, it has been found in some studies that transverse colon cancer has different biological characteristics from RCC [15,16,17,18]. Therefore, the conflicting results may be partly due to the inconsistency of location grouping criteria, and the dichotomy model (right-sided vs. left-sided colon) may be inappropriate for the study of colon cancer [16,17,18,19]. Moreover, the prognostic differences between specific subsites (cecum, ascending colon, hepatic flexure, transverse colon, splenic flexure, descending colon, and sigmoid colon) have also been reported in a few studies [19,20,21]. In order to further explore the effects of tumor locations on prognosis, it is essential to find two parts with the greatest survival differences by comparing different primary tumor locations. 

In our study, we collected the data of patients with stage I–III colon cancer from the Surveillance, Epidemiology, and End Results (SEER) Program database. By analyzing colon cancer data in the SEER database, it was found that the survival difference between cecum and sigmoid colon cancer was most significant among different locations. Then the prognostic significance of the two locations was further explored in multivariate survival analyses after PSM. 

## 2. Materials and Methods

### 2.1. Data Sources

Colon cancer data are from the SEER database. SEER is an authoritative source for cancer statistics, consisting of 18 population-based cancer registries and covering approximately 34.6% of patients with cancer in the United States. Clinicopathological information, including patient demographics, primary tumor location, tumor morphology and stage at diagnosis, treatment course, and follow-up for vital status, was extracted using the “case listing” option of the SEER*Stat 8.3.6 software. 

### 2.2. Study Design

Figure 1 shows the flowchart of the whole study design. According to chronological order, data of colon cancer between 2004 and 2015 in SEER were divided into two groups (2004–2009 and 2010–2015) with December 2009 as the cut-off point. Cancer-specific survival (CSS) was defined as an endpoint of death from cause relevant with colon cancer. Overall survival (OS) was defined as an endpoint of death from any cause, whichever came first. By using univariate Cox regression analyses in both 2004–2009 and 2010–2015, two tumor locations with the most significant survival difference were explored. Then PSM method was carried out between those two tumor locations in the two groups, respectively. Finally, survival analyses were performed to further validate the prognostic significance of the two tumor locations with the most significant survival difference.

### 2.3. Patients Selection

Primary selection criteria for study cases included: (1) diagnosis of colon cancer between 2004 and 2015; (2) histological confirmation of colon adenocarcinoma; (3) definite tumor location on colon, excluding transverse colon (since it is still difficult to strictly define the scope of the right-sided or left-sided colon, it is not appropriate to simply classify transverse colon as which side of the colon); (4) survival time known; (5) suffering from colon cancer only; (6) radical surgery performed; (7) number of lymphadenectomy and lymph node metastasis known; (8) no distant metastasis and American Joint Committee on Cancer (AJCC) T1–T4; (9) differentiated grade known; (10) age range: 18–80 years old. A total of 92,005 patients with colon cancer in SEER met our inclusion criteria (Figure 2). 

### 2.4. Statistical Analyses

Frequency distribution of clinicopathological variables was compared with the χ^2^ test. Survival curves were generated using the Kaplan–Meier method and were compared using the log-rank test. Univariate and multivariate analyses were performed using Cox proportional hazards model. The variables with *p* < 0.1 in the univariate analyses were included in a multivariate proportional hazard regression model. To further minimize biases in baseline characteristics, the PSM method was adopted to balance covariates with statistical differences between cecum and sigmoid colon groups. The patients in the cecum group were then matched as 1:1 with those in the sigmoid colon group by nearest neighbor matching. Standardized mean difference (SMD) was used to weigh the biases of each variable between the two groups. A caliper was used to define the maximum allowable difference (SMD < 0.1) between two groups to ensure good matches. All tests were two-sided, and *p* < 0.05 was considered statistically significant. Statistical analyses were performed using SPSS 20.0 and R software 3.6.3. 

## 3. Results

### 3.1. Demographic and Clinicopathological Features in Total Cohort

A total of 92,005 patients diagnosed with colon cancer between 2004 and 2015 were included in the study. The median follow-up time was 56 months (IQR 27 to 96 months) in overall cohort. At the end of the follow-up period, 66,224 (71.98%) patients were alive, 15,010 (16.31%) died from colon cancer, and 15,010 (11.71%) died due to other reasons. These patients were divided into two groups according to the time of diagnosis. Among all patients, 45,645 (49.6%) patients were diagnosed between 2004 and 2009, and the remaining 46,360 (50.4%) patients were diagnosed between 2010 and 2015 (Appendix A). 

Univariate regression COX analyses showed that patients with RCC had worse CSS (HR [95%CI]: 1.09 [1.04–1.13] for 2004–2009; HR [95% CI]: 1.20 [1.14–1.27] for 2010–2015) and OS (HR [95%CI]: 1.16 [1.13–1.20] for 2004–2009; HR [95% CI]: 1.22 [1.17–1.28] for 2010–2015) compared to LCC. 

In order to further analyze the greatest survival differences between RCC and LCC, we separated right-sided and left-sided cancer into three parts respectively: cecum (2004–2009: 16,313/45,645, 35.7%; 2010–2015: 16,175/46,360, 34.9%), ascending colon (2004–2009: 3308/45,645, 7.2%; 2010–2015: 3568/46,360, 7.7%), hepatic flexure (2004–2009: 1961/45,645, 4.3%; 2010–2015: 1798/46,360, 3.9%), splenic flexure (2004–2009: 2637/45,645, 5.8%; 2010–2015: 2348/46,360, 5.1%), descending colon (2004–2009: 9935/45,645, 21.8%; 2010–2015: 10,941/46,360, 23.6%), and sigmoid colon (2004–2009: 11,491/45,645, 25.2%; 2010–2015: 11,530/46,360, 24.9%) (Appendix A). 

In univariate Cox regression analysis, when sigmoid colon cancer was compared to colon cancer in other locations, we found that CSS difference between patients with cecum cancer and patients with sigmoid colon cancer (HR [95% CI]: 1.24 [1.18–1.30]) was second only to that between splenic flexure colon cancer and sigmoid colon cancer (HR [95% CI]: 1.26 [1.14–1.38]) (Figure 3A), while OS difference between patients with cecum cancer and patients with sigmoid colon cancer (HR [95% CI]: 1.25 [1.21–1.30]) was the most significant in the 2004–2009 group (Figure 3B) (Table 1). In the 2010–2015 group, among six locations, patients with cecum cancer and sigmoid colon cancer had the most remarkable survival difference in both CSS (HR [95% CI]: 1.39 [1.30–1.49]; Figure 3C) and OS (HR [95% CI]: 1.35 [1.28–1.43]; Figure 3D) (Table 1).

### 3.2. Clinicopathological Features for Cecum Cancer Compared with Sigmoid Colon Cancer

The demographics and clinicopathological features of patients with cecum cancer versus sigmoid colon cancer are summarized in Table 2. In the 2004–2009 group, the distribution of clinicopathological variables in both tumor locations showed significant difference except for lymph node metastasis (LNM) (*p* = 0.507). In the 2010–2015 group, all other variables, not including LNM (*p* = 0.615) and CEA (*p* = 0.125), presented an obvious imbalance between cecum and sigmoid colon cancer. Patients with cecum cancer were more likely to be older (*p* < 0.001 for both groups), female (*p* < 0.001 for both groups), and Caucasian (*p* < 0.001 for both groups) than those with sigmoid colon cancer. Compared with sigmoid colon cancer, patients with cecum cancer accounted for a larger proportion in poor differentiation (*p* < 0.001 for both groups), more lymph node dissection (LND) ≥ 12 (*p* < 0.001 for both groups), late TNM stage (*p* < 0.001 for both groups), positive CEA value (*p* = 0.003 for the 2004–2009 group), and more LNM (*p* = 0.003 for the 2010–2015 group). In addition, fewer patients diagnosed with cecum cancer received radiotherapy (*p* < 0.001 for both groups) or chemotherapy (*p* < 0.001 for both groups) in contrast to those with sigmoid colon cancer.

To adjust for these differences in baseline characteristics and minimize biases between both primary cancer locations, a PSM analysis was performed (Table 3). After matching with 1:1 ratio for cecum and sigmoid colon cancer, there were 9926 patients in each tumor location for the 2004–2009 group, and 10,044 patients in each tumor location were included in the final analysis for the 2010–2015 group. Appendix A showed that matched results are basically consistent with the weighted data (SMD < 0.1). 

### 3.3. Survival Analyses after PSM for Patients Diagnosed with Cecum or Sigmoid Cancer

In the univariate Cox regression analyses, a worse survival outcome was found in patients with cecum cancer compared to those with sigmoid colon cancer for both 2004–2009 (HR [95% CI]: 1.08 [1.02–1.14] for CSS; HR [95% CI]: 1.07 [1.02–1.11] for OS) and 2010–2015 groups (HR [95% CI]: 1.23 [1.14–1.33] for CSS; HR [95% CI]: 1.19 [1.12–1.27] for OS) (Figure 4; Table 4).

Table 5 presents the multivariate Cox regression analyses of prognostic factors after PSM. For the 2004–2009 group, the multivariate analyses identified cecum cancer as an independent unfavorable factor of CSS (HR [95% CI]: 1.11 [1.04–1.17]) and OS (HR [95% CI]: 1.09 [1.04–1.14]) relative to sigmoid colon cancer. Similarly, we found that patients with cecum cancer had a worse CSS (HR [95% CI]: 1.23 [1.13–1.33]) and OS (HR [95% CI]: 1.19 [1.11–1.27]) than cecum cancer in the 2010–2015 group.

## 4. Discussion

Right-sided and left-sided colons deriving from different embryologic origin are clinically and molecularly distinct. However, the influence of primary tumor locations on the prognosis remains controversial [20,22,23,24,25]. In the present study, we found that patients with RCC had worse CSS and OS than LCC, and the two tumor locations with the greatest survival difference were cecum and sigmoid colon. Moreover, multivariate Cox regression analyses further demonstrated the worse prognosis (CSS and OS) of patients with cecum cancer compared with sigmoid colon cancer after PSM, which is helpful to further explore the influence of primary tumor locations on prognosis. To the best of our knowledge, this is the first study to find that the prognostic difference between cecum and sigmoid colon cancer is the greatest between RCC and LCC, which may partly account for the prognostic difference between RCC and LCC.

Currently, most of previous studies demonstrated a poorer prognosis in RCC versus LCC, which is consistent with the present study [1,2,3,4,6,7]. A large retrospective study presented a lower five-year survival rate for RCC (70.4%) relative to LCC (74.0%) in Japan [1]. And RCC also showed a significantly worse five-year survival (HR [95% CI]: 1.71 [1.10–2.64], *p* = 0.017) in a retrospective study of patients with colon cancer present in the Cancer Genome Atlas (TCGA) [14]. By analyzing the prognosis of patients with unresectable colon cancer liver metastasis, Zhao et al. found that the risk of survival deterioration of RCC was significantly higher than that of LCC [26]. A meta-analysis from 66 researches reported that patients with RCC had their risk of death increased by 18% compared to LCC, which was independent of stage [7]. Another recent meta-analysis from 14 studies on metastatic colorectal cancer reported that primary cancer originating from the right-sided colon was significantly related with a worse survival in contrast to left-sided colon [3]. In addition, the present study divided the data into two parts according to the time period so as to reduce the confounding bias caused by the large time span. The difference in the prognosis of patients with RCC and LCC was shown to be consistent in the two time groups, which further confirmed the worse prognosis of RCC over LCC.

As a large sample database with longitudinal data, it is very suitable to utilize the data from the SEER database to analyze the prognostic significance of primary tumor locations. Although there have been several studies about the survival difference between RCC and LCC based on the SEER database, few studies further analyzed the prognostic differences of more precise tumor locations [8,9,10,11,19,27]. The study published by Meguid et al. including 77,978 patients who underwent surgical resection for aggressive colon cancer from 1988 to 2003 in the SEER database showed a 4.2% increased mortality risk associated with RCC versus LCC [8]. Furthermore, the subset analyses stratified by AJCC stage revealed that the higher mortality risk of patients with RCC was observed in stages III and IV compared to patients with LCC [8]. Weiss et al. summarized the data of colon cancer from 1992 to 2005. Although there was no statistical difference in the prognosis between patients with RCC and LCC in the overall cohort, further stratified analysis indicated a higher mortality of patients with RCC compared to LCC in stage III [10]. In order to further explore the prognostic significance between RCC and LCC, Warschkow et al. collected the data of patients with stage I–III colon cancer from the SEER database between 2004 and 2012 and performed a PSM analysis to minimize biases between both primary cancer locations [11]. After PSM, the survival prognosis of patients with RCC was found to be superior to those with LCC regarding OS and CSS in overall cohort, which contradicts our study [11]. However, although Warschkow et al. adopted the PSM, they did not adjust for radiotherapy and chemotherapy in baseline characteristics, which obviously has a great impact on the survival analyses [11]. In addition, our study not only summarized more recent data from 2005 to 2015, but also divided the data into two groups (2004–2009 and 2010–2015) based on time intervals for mutual verification between the two time periods, which reduced the bias due to inconsistent previous treatment standards. Wang et al. also made full use of the data from the SEER database to analyze the survival distinction between RCC and LCC by adopting a competing risk model. The results showed that the cancer-specific mortality (CSM) of RCC significantly increased compared with LCC in the overall cohort [9]. Obviously, not only the inclusion and exclusion criteria of these studies based on the SEER database were inconsistent, but also these grouping criteria according to primary tumor locations were different [8,9,10,11]. Moreover, these studies did not further explore the prognostic differences among more precise tumor locations, especially cecum and sigmoid colon [8,9,10,11]. 

The present study showed that the survival difference between cecum cancers and sigmoid colon cancers is the largest among the six colon sites, which is akin to the results of Shaib et al. [21]. By analyzing the data of patients with non-metastatic, invasive right-sided adenocarcinoma of the colon from 1988 to 2014 who underwent partial colectomy in SEER, Nasseri at al found that cecum cancers were prone to poorer disease-specific survival (median 86.0, 93.0, and 89.0 months, respectively, *p* < 0.001) and OS (median not reached, *p* < 0.001) compared with cancers in other sites of the right colon [19]. In addition, Ben-Aharon et al. also found that the Oncotype Recurrence Score, a clinically validated predictor of recurrence risk in patients with stage II CRC, gradually decreased across the colon (cecum, highest score; sigmoid, lowest score; *p* = 0.04) [20]. In the present study, patients with cecum cancer had more poorly differentiated tumor, advanced LNM, and late TNM stage compared with sigmoid colon cancer, which is in line with results of previous studies [1,2,9,28]. The prognosis of cecum cancer was worse than sigmoid colon cancer regardless of 2004–2009 or 2010–2015 group in univariate analyses. It is worth noting that the prognostic difference between cecum cancer and sigmoid colon cancer is still significant after PSM. Moreover, after adjusting for differences in clinicopathological characteristics, the multivariate analyses identified cecum cancer as an independent unfavorable factor of CSS and OS relative to sigmoid colon cancer. All in all, our study further confirmed that the location of primary colon cancer was an important prognostic factor and suggested that the prognosis of cecum is worse than that of sigmoid colon cancer.

The reasons might be related to the different embryological origins of colon tissue—proximal colon (right-sided) deriving from mid-gut and distal colon (left-sided) deriving from hind-gut [29]. Meanwhile, different gut microbiota in left and right-sided colon cause differences in colonic mucosal immunology, which theoretically should be the largest between the cecum and the sigmoid colon [30]. From the perspective of clinical symptoms, obstruction and hematochezia due to LCC occurred more frequently than those due to RCC, which is helpful for the early diagnosis and treatment of sigmoid colon cancer [31]. Ward et al. hypothesized three pathways through which tumor site impacts survival differences, including TNM stage, microsatellite instability (MSI), and other genetic drivers [14]. Many previous studies have suggested that the potential molecular mechanism contributing to the different prognosis between RCC and LCC might be associated with differences in gene expression and signaling pathways [14,20,28,32,33,34,35,36]. Some chemotherapy regimens were also found to have different efficacy between LCC and RCC due to different microsatellite status [34]. Anatomically, the difference between cecum and sigmoid colon is the largest, which may suggest that the prognostic difference between cecum and sigmoid colon cancer is also most significant. In addition, the mobility of the sigmoid colon is better than that of the cecum, which helps to improve the curative effect of surgery and survival.

There are several limitations in the present study. First, although this study grouped the cases according to the time interval and analyzed them separately, due to the large time span, it is still difficult to eliminate the influence of inconsistent treatment standards before and after on the study results. Second, a few confounders are not matched perfectly despite adopting PSM method, which may influence the results to some extent. Third, other potential biases from unobserved confounders may be ignored, such as MSI status, socioeconomic status, environmental exposures, and so on.

## 5. Conclusions

The present study indicates that the prognosis of patients with cecum cancer is worse than those with sigmoid colon cancer, which may partly account for the prognostic difference between RCC and LCC. The dichotomy model (right-sided vs. left-sided colon) may be inappropriate for the study of colon cancer. Tumor locations should be further refined in future research and treatment so as to provide better guidance for individualized treatment of patients with colon cancer. Further studies are very worthy to analyze the potential molecular mechanism and instruct individualized treatment.

## Figures and Tables

**Figure 1 medicina-59-00045-f001:**
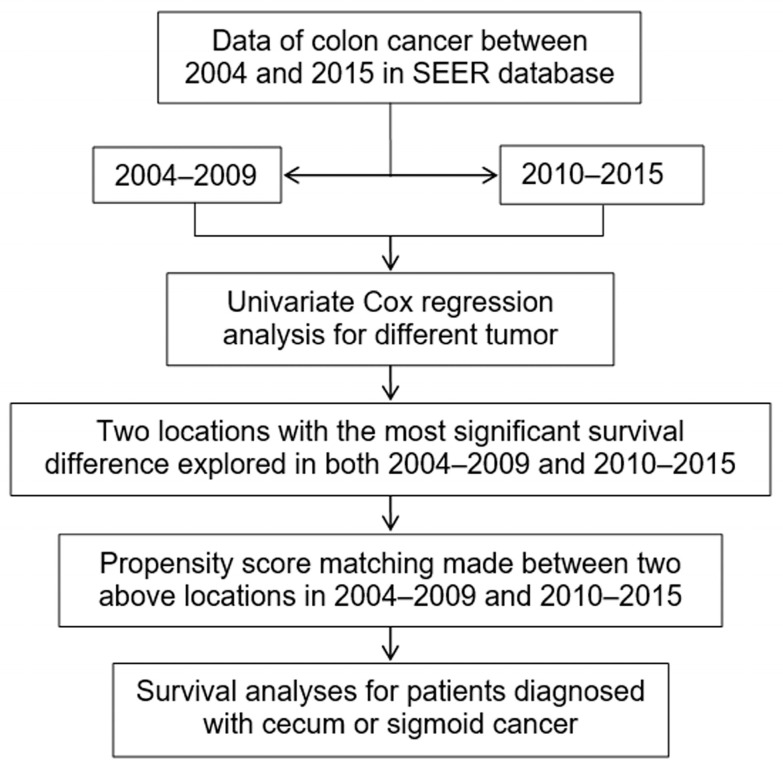
Research approach for colon cancer. Footnotes: S*EER*, the Surveillance, Epidemiology, and End Results.

**Figure 2 medicina-59-00045-f002:**
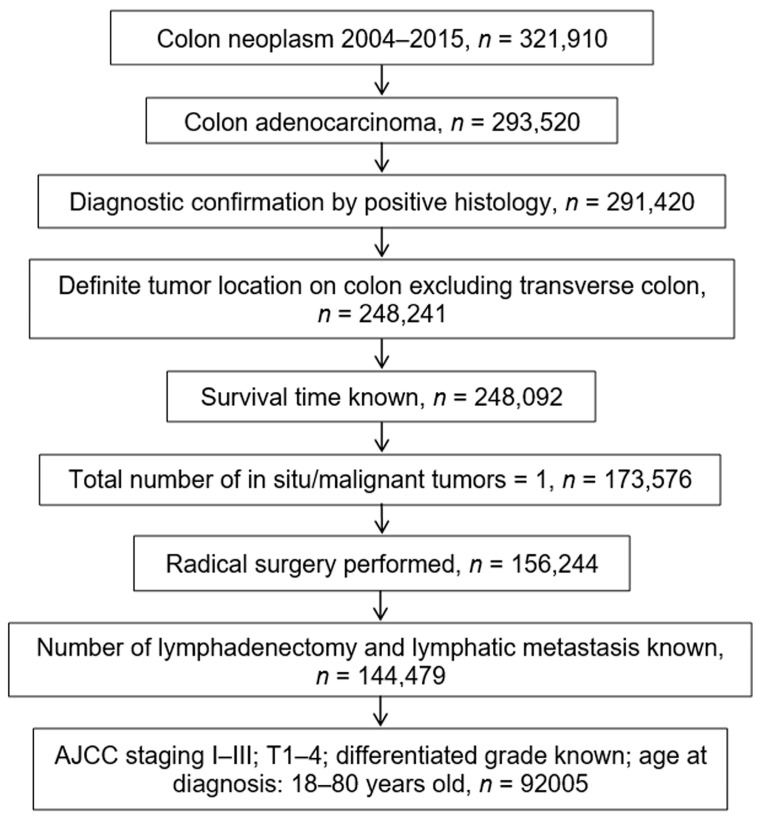
Screening flowchart for colon cancer in the SEER database. Footnotes: *AJCC*, American Joint Committee on Cancer; *T*, primary tumor; *SEER*, the Surveillance, Epidemiology, and End Results.

**Figure 3 medicina-59-00045-f003:**
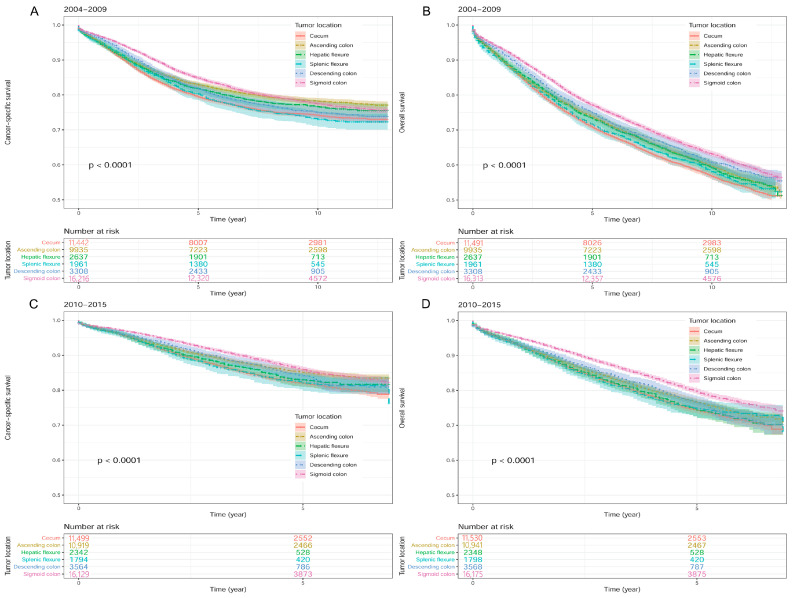
Kaplan-Meier survival curves according to different colon cancer locations in the SEER database. (**A**) Cancer-specific survival in 2004–2009. (**B**) Overall survival in 2004–2009. (**C**) Cancer-specific survival in 2010–2015. (**D**) Overall survival in 2010–2015. Footnotes: S*EER*, the Surveillance, Epidemiology, and End Results.

**Figure 4 medicina-59-00045-f004:**
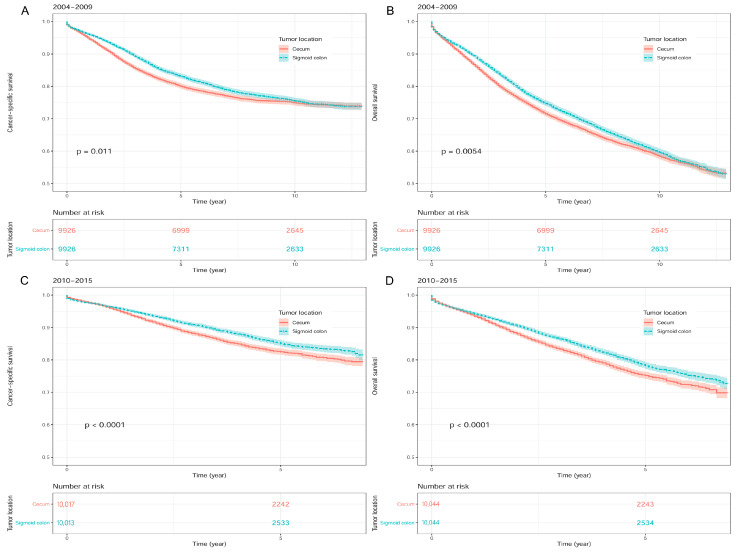
Kaplan-Meier survival curves between sigmoid colon cancer and cecum cancer after propensity score matching in SEER database. (**A**) Cancer-specific survival in 2004–2009. (**B**) Overall survival in 2004–2009. (**C**) Cancer-specific survival in 2010–2015. (**D**) Overall survival in 2010–2015. Footnotes: S*EER*, the Surveillance, Epidemiology, and End Results.

**Table 1 medicina-59-00045-t001:** Univariate Cox regression analysis of different tumor locations in patients with colon cancer in the SEER database.

Tumor Location	2004–2009		2010–2015	
	CSS	OS	CSS	OS
	HR (95% CI)	HR (95% CI)	HR (95% CI)	HR (95% CI)
Classification 1				
LCC	Ref.	Ref.	Ref.	Ref.
RCC	1.09 (1.04–1.13)	1.16 (1.13–1.20)	1.20 (1.14–1.27)	1.22 (1.17–1.28)
Classification 2				
Sigmoid colon	Ref.	Ref.	Ref.	Ref.
Descending colon	1.15 (1.06–1.24)	1.08 (1.01–1.15)	1.19 (1.06–1.32)	1.23 (1.12–1.34)
Splenic flexure	1.26 (1.14–1.38)	1.20 (1.12–1.30)	1.37 (1.19–1.57)	1.32 (1.18–1.48)
Hepatic flexure	1.10 (1.01–1.21)	1.15 (1.08–1.23)	1.29 (1.14–1.46)	1.33 (1.20–1.47)
Ascending colon	1.02 (0.96–1.08)	1.15 (1.10–1.19)	1.14 (1.06–1.23)	1.24 (1.17–1.31)
Cecum	1.24 (1.18–1.30)	1.25 (1.21–1.30)	1.39 (1.30–1.49)	1.35 (1.28–1.43)

*LCC*, left-sided colon cancer; *RCC*, right-sided colon cancer; *SEER*, the Surveillance, Epidemiology, and End Results; *CSS*, cancer-specific survival; *OS*, overall survival; *HR*, hazard ratio; *CI*, confidence interval.

**Table 2 medicina-59-00045-t002:** Comparison of clinicopathological data of patients with sigmoid colon cancer versus cecum cancer in the SEER database.

Variable	2004–2009			2010–2015		
	Sigmoid Colon	Cecum	*p*-Value	Sigmoid Colon	Cecum	*p*-Value
No. subjects (%)	16,313 (58.7)	11,491 (41.3)		16,175 (58.4)	11,530 (41.6)	
Age (years)						
≤Median age	9848 (60.4)	5098 (44.4)		9980 (61.7)	5135 (44.5)	
>Median age	6465 (39.6)	6393 (55.6)	<0.001	6195 (38.3)	6395 (55.5)	<0.001
Sex						
Female	7615 (46.7)	6176 (53.7)		7326 (45.3)	5986 (51.9)	
Male	8698 (53.3)	5315 (46.3)	<0.001	8849 (54.7)	5544 (48.1)	<0.001
Ethnicity						
Caucasian	12,699 (77.8)	9178 (79.9)		12,229 (75.6)	9023 (78.3)	
African-American	1662 (10.2)	1598 (13.9)		1744 (10.8)	1677 (14.5)	
Others/Unknown	1952 (12.0)	715 (6.2)	<0.001	2202 (13.6)	830 (7.2)	<0.001
Differentiation						
Well	1624 (10.0)	1128 (9.8)		1545 (9.6)	1060 (9.2)	
Moderately	12,655 (77.6)	7884 (68.6)		12,778 (79.0)	8192 (71.0)	
Poorly	1899 (11.6)	2292 (19.9)		1594 (9.9)	1867 (16.2)	
Undifferentiated	135 (0.8)	187 (1.6)	<0.001	258 (1.6)	411 (3.6)	<0.001
LND						
<12	6512 (39.9)	2686 (23.4)		2913 (18.0)	944 (8.2)	
≥12	9801 (60.1)	8805 (76.6)	<0.001	13,262 (82.0)	10,586 (91.8)	<0.001
LNM						
No	9979 (61.2)	6984 (60.8)		9936 (61.4)	7117 (61.7)	
Yes	6334 (38.8)	4507 (39.2)	0.507	6239 (38.6)	4413 (38.3)	0.615
AJCC T-stage						
T1	3140 (19.2)	1243 (10.8)		3169 (19.6)	1508 (13.1)	
T2	2669 (16.4)	2209 (19.2)		2614 (16.2)	2229 (19.3)	
T3	8841 (54.2)	6463 (56.2)		8206 (50.7)	5792 (50.2)	
T4	1663 (10.2)	1576 (13.7)	<0.001	2186 (13.5)	2001 (17.4)	<0.001
AJCC N-stage						
N0	9931 (60.9)	6951 (60.5)		9615 (59.4)	6969 (60.4)	
N1 ^†^	4195 (25.7)	2699 (23.5)		4520 (27.9)	2783 (24.1)	
N2	2187 (13.4)	1841 (16.0)	<0.001	2040 (12.6)	1778 (15.4)	<0.001
AJCC TNM stage						
I	4681 (28.7)	2926 (25.5)		4625 (28.6)	3089 (26.8)	
II	5250 (32.2)	4025 (35.0)		4990 (30.9)	3880 (33.7)	
III	6382 (39.1)	4540 (39.5)	<0.001	6560 (40.6)	4561 (39.6)	<0.001
CEA						
Negative	6038 (37.0)	4255 (37.0)		6263 (38.7)	4486 (38.9)	
Positive	2943 (18.0)	2260 (19.7)		3138 (19.4)	2318 (20.1)	
Borderline	68 (0.4)	51 (0.4)		47 (0.3)	46 (0.4)	
Others	7264 (44.5)	4925 (42.9)	0.003	6727 (41.6)	4680 (40.6)	0.125
Radiation therapy						
No	15,742 (96.5)	11,304 (98.4)		15,716 (97.2)	11,402 (98.9)	
Yes	571 (3.5)	187 (1.6)	<0.001	459 (2.8)	128 (1.1)	<0.001
Chemotherapy						
No/unknown	10402 (63.8)	7588 (66.0)		10,090 (62.4)	7468 (64.8)	
Yes	5911 (36.2)	3903 (34.0)	<0.001	6085 (37.6)	4062 (35.2)	<0.001

S*EER*, the Surveillance, Epidemiology, and End Results; *LND*, lymph node dissection; *LNM*, lymph node metastasis; *AJCC*, American Joint Committee on Cancer; *T*, primary tumor; *N*, regional lymph nodes; *M*, distant metastasis; *CEA*, carcinoembryonic antigen; ^†^ N1, included tumor deposits besides lymph node metastasis.

**Table 3 medicina-59-00045-t003:** Comparison of clinicopathological features of patients with sigmoid colon cancer versus cecum cancer after propensity score matching in the SEER database.

Variable	2004–2009			2010–2015		
	Sigmoid Colon	Cecum	*p*-Value	Sigmoid Colon	Cecum	*p*-Value
No. subjects (%)	9926 (50.0)	9926 (50.0)		10,044 (50.0)	10,044 (50.0)	
Age (years)						
≤Median age	4837 (48.7)	4943 (49.8)	0.136	4991 (49.7)	5071 (50.5)	0.265
>Median age	5089 (51.3)	4983 (50.2)		5053 (50.3)	4973 (49.5)	
Sex						
Female	5017 (50.5)	4993 (50.3)		4915 (48.9)	4893 (48.7)	
Male	4909 (49.5)	4933 (49.7)	0.744	5129 (51.1)	5151 (51.3)	0.767
Ethnicity						
Caucasian	7838 (79.0)	7912 (79.7)		7763 (77.3)	7819 (77.8)	
African-American	1151 (11.6)	1311 (13.2)		1216 (12.1)	1404 (14.0)	
Others/Unknown	937 (9.4)	703 (7.1)	<0.001	1065 (10.6)	821 (8.2)	<0.001
Differentiation						
Well	997 (10.0)	996 (10.0)		937 (9.3)	941 (9.4)	
Moderately	7254 (73.1)	7266 (73.2)		7538 (75.0)	7500 (74.7)	
Poorly	1556 (15.7)	1548 (15.6)		1335 (13.3)	1365 (13.6)	
Undifferentiated	119 (1.2)	116 (1.2)	0.995	234 (2.3)	238 (2.4)	0.925
LND						
<12	2580 (26.0)	2641 (26.6)		832 (8.3)	941 (9.4)	
≥12	7346 (74.0)	7285 (73.4)	0.333	9212 (91.7)	9103 (90.6)	0.007
LNM						
No	6044 (60.9)	6027 (60.7)		6220 (61.9)	6176 (61.5)	
Yes	3882 (39.1)	3899 (39.3)	0.816	3824 (38.1)	3868 (38.5)	0.533
AJCC T-stage						
T1	1234 (12.4)	1217 (12.3)		1495 (14.9)	1456 (14.5)	
T2	1818 (18.3)	1829 (18.4)		1830 (18.2)	1793 (17.9)	
T3	5644 (56.9)	5675 (57.2)		5129 (51.1)	5190 (51.7)	
T4	1230 (12.4)	1205 (12.1)	0.921	1590 (15.8)	1605 (16.0)	0.723
AJCC N-stage						
N0	6022 (60.7)	5996 (60.4)		6079 (60.5)	6031 (60.0)	
N1 ^†^	2404 (24.2)	2400 (24.2)		2512 (25.0)	2549 (25.4)	
N2	1500 (15.1)	1530 (15.4)	0.837	1453 (14.5)	1464 (14.6)	0.778
AJCC TNM stage						
I	2561 (25.8)	2535 (25.5)		2738 (27.3)	2672 (26.6)	
II	3461 (34.9)	3461 (34.9)		3341 (33.3)	3359 (33.4)	
III	3904 (39.3)	3930 (39.6)	0.896	3965 (39.5)	4013 (40.0)	0.565
CEA						
Negative	3756 (37.8)	3732 (37.6)		3915 (39.0)	3913 (39.0)	
Positive	1866 (18.8)	1910 (19.2)		2043 (20.3)	1964 (19.6)	
Borderline	41 (0.4)	44 (0.4)		23 (0.2)	42 (0.4)	
Others	4263 (42.9)	4240 (42.7)	0.860	4063 (40.5)	4125 (41.1)	0.056
Radiation therapy						
No	9758 (98.3)	9743 (98.2)		9939 (99.0)	9916 (98.7)	
Yes	168 (1.7)	183 (1.8)	0.451	105 (1.0)	128 (1.3)	0.147
Chemotherapy						
No/unknown	6456 (65.0)	6479 (65.3)		6471 (64.4)	6384 (63.6)	
Yes	3470 (35.0)	3447 (34.7)	0.743	3573 (35.6)	3660 (36.4)	0.206

Footnotes: *SEER*, the Surveillance, Epidemiology, and End Results; *LND*, lymph node dissection; *LNM*, lymph node metastasis; *AJCC*, American Joint Committee on Cancer; *T*, primary tumor; *N*, regional lymph nodes; *M*, distant metastasis; *CEA*, carcinoembryonic antigen; ^†^ N1 included tumor deposits besides lymph node metastasis.

**Table 4 medicina-59-00045-t004:** Univariate Cox regression analysis of prognostic factors after propensity score matching for patients diagnosed with cecum or sigmoid cancer.

Variable	2004–2009					2010–2015				
	n	CSS		OS		n	CSS		OS	
		HR (95% CI)	*p*	HR (95% CI)	*p*		HR (95% CI)	*p*	HR (95% CI)	*p*
Tumor location										
Sigmoid colon	9926	Ref.		Ref.		10,044	Ref.		Ref.	
Cecum	9926	1.08 (1.02–1.14)	0.011	1.07 (1.02–1.11)	0.005	10,044	1.23 (1.14–1.33)	<0.001	1.19 (1.12–1.27)	<0.001
Age (years)										
≤Median age	9780	Ref.		Ref.		10,062	Ref.		Ref.	
>Median age	10,072	1.37 (1.29–1.45)	<0.001	2.16 (2.06–2.26)	<0.001	10,026	1.39 (1.28–1.50)	<0.001	1.81 (1.69–1.93)	<0.001
Sex										
Female	10,010	Ref.		Ref.		9808	Ref.		Ref.	
Male	9842	1.11 (1.04–1.17)	<0.001	1.20 (1.15–1.26)	<0.001	10,280	1.05 (0.97–1.14)	0.207	1.19 (1.12–1.27)	<0.001
Ethnicity										
Caucasian	15,750	Ref.		Ref.		15,582	Ref.		Ref.	
African-American	2462	1.37 (1.26–1.49)	<0.001	1.19 (1.12–1.27)	<0.001	2620	1.38 (1.24–1.54)	<0.001	1.30 (1.19–1.42)	<0.001
Others/Unknown	1640	0.85 (0.75–0.95)	0.005	0.80 (0.73–0.87)	<0.001	1886	0.90 (0.78–1.04)	0.165	0.84 (0.74–0.95)	0.005
Differentiation										
Well	1993	Ref.		Ref.		1878	Ref.		Ref.	
Moderately	14,520	1.57 (1.39–1.77)	<0.001	1.26 (1.16–1.37)	<0.001	15,038	1.54 (1.29–1.84)	<0.001	1.31 (1.15–1.49)	<0.001
Poorly	3104	2.65 (2.32–3.02)	<0.001	1.68 (1.53–1.84)	<0.001	2700	3.11 (2.57–3.76)	<0.001	2.10 (1.81–2.43)	<0.001
Undifferentiated	235	2.39 (1.84–3.10)	<0.001	1.72 (1.41–2.11)	<0.001	472	3.93 (3.06–5.05)	<0.001	2.57 (2.09–3.16)	<0.001
LND										
<12	5221	Ref.		Ref.		1773	Ref.		Ref.	
≥12	14,631	0.80 (0.75–0.85)	<0.001	0.73 (0.70–0.77)	<0.001	18,315	0.65 (0.58–0.73)	<0.001	0.63 (0.58–0.70)	<0.001
LNM										
No	12,071	Ref.		Ref.		12,396	Ref.		Ref.	
Yes	7781	3.16 (2.97–3.36)	<0.001	1.79 (1.71–1.87)	<0.001	7692	3.35 (3.08–3.64)	<0.001	2.21 (2.07–2.36)	<0.001
AJCC T-stage										
T1	2451	Ref.		Ref.		2951	Ref.		Ref.	
T2	3647	1.41 (1.17–1.70)	<0.001	1.19 (1.07–1.31)	<0.001	3623	1.85 (1.38–2.49)	<0.001	1.46 (1.23–1.73)	<0.001
T3	11,319	4.38 (3.73–5.14)	<0.001	1.91 (1.76–2.08)	<0.001	10,319	5.91 (4.60–7.60)	<0.001	2.70 (2.34–3.11)	<0.001
T4	2435	10.01 (8.48–11.82)	<0.001	3.53 (3.21–3.88)	<0.001	3195	16.90 (13.12–21.77)	<0.001	5.87 (5.07–6.80)	<0.001
AJCC N-stage										
N0	12,018	Ref.		Ref.		12,110	Ref.		Ref.	
N1 ^†^	4804	2.38 (2.22–2.56)	<0.001	1.48 (1.41–1.56)	<0.001	5061	2.51 (2.27–2.77)	<0.001	1.80 (1.67–1.95)	<0.001
N2	3030	4,57 (4.26–4.91)	<0.001	2.36 (2.23–2.49)	<0.001	2917	5.34 (4.84–5.88)	<0.001	3.17 (2.93–3.44)	<0.001
AJCC TNM stage										
I	5096	Ref.		Ref.		5410	Ref.		Ref.	
II	6922	3.13 (2.78–3.54)	<0.001	1.58 (1.48–1.69)	<0.001	6700	4.07 (3.38–4.90)	<0.001	2.02 (1.81–2.25)	<0.001
III	7834	6.87 (6.13–7.70)	<0.001	2.36 (2.21–2.51)	<0.001	7978	9.25 (7.75–11.04)	<0.001	3.53 (3.18–3.91)	<0.001
CEA										
Negative	7488	Ref.		Ref.		7828	Ref.		Ref.	
Positive	3776	2.19 (2.03–2.36)	<0.001	1.91 (1.80–2.03)	<0.001	4007	2.61 (2.35–2.90)	<0.001	2.15 (1.97–2.34)	<0.001
Borderline	85	1.68 (1.11–2.54)	0.014	1.66 (1.22–2.26)	0.001	65	1.70 (0.88–3.29)	0.113	1.39 (0.79–2.46)	0.255
Others	8503	1.31 (1.22–1.41)	<0.001	1.35 (1.28–1.42)	<0.001	8188	1.53 (1.38–1.69)	<0.001	1.46 (1.35–1.58)	<0.001
Radiation therapy										
No	19,501	Ref.		Ref.		19,855	Ref.		Ref.	
Yes	351	2.26 (1.92–2.65)	<0.001	1.62 (1.41–1.87)	<0.001	233	2.33 (1.83–2.97)	<0.001	1.78 (1.42–2.22)	<0.001
Chemotherapy										
No/unknown	12,935	Ref.		Ref.		12,855	Ref.		Ref.	
Yes	6917	1.67 (1.57–1.77)	<0.001	1.04 (0.99–1.09)	0.128	7233	2.33 (1.83–2.97)	<0.001	1.78 (1.42–2.22)	<0.001

*SEER*, the Surveillance, Epidemiology, and End Results; *LND*, lymph node dissection; *LNM*, lymph node metastasis; *AJCC*, American Joint Committee on Cancer; *T*, primary tumor; *N*, regional lymph nodes; *M*, distant metastasis; *CEA*, carcinoembryonic antigen; *CSS*, cancer-specific survival; *OS*, overall survival; *HR*, hazard ratio; *CI*, confidence interval; ^†^ N1 included tumor deposits besides lymph node metastasis.

**Table 5 medicina-59-00045-t005:** Multivariate Cox regression analysis of prognostic factors after propensity score matching. Variables with *p* value less than 0.01 in univariate analysis were incorporated into multivariate Cox regression analysis.

Variable	2004–2009					2010–2015				
	n	CSS		OS		n	CSS		OS	
		HR (95% CI)	*p*	HR (95% CI)	*p*		HR (95% CI)	*p*	HR (95% CI)	*p*
Tumor location										
Sigmoid colon	9926	Ref.		Ref.		10,044	Ref.		Ref.	
Cecum	9926	1.11 (1.04–1.17)	0.001	1.09 (1.04–1.14)	<0.001	10,044	1.23 (1.13–1.33)	<0.001	1.19 (1.11–1.27)	<0.001
Age (years)										
≤Median age	9780	Ref.		Ref.		10,062	Ref.		Ref.	
>Median age	10,072	1.53 (1.44–1.63)	<0.001	2.22 (2.12–2.33)	<0.001	10,026	1.59 (1.47–1.73)	<0.001	1.92 (1.79–2.06)	<0.001
Sex										
Female	10,010	Ref.		Ref.		9808	Ref.		Ref.	
Male	9842	1.14 (1.07–1.21)	<0.001	1.26 (1.20–1.31)	<0.001	10,280	-	-	1.19 (1.11–1.27)	<0.001
Ethnicity										
Caucasian	15,750	Ref.		Ref.		15,582	Ref.		Ref.	
African-American	2462	1.43 (1.31–1.55)	<0.001	1.32 (1.24–1.41)	<0.001	2620	1.44 (1.29–1.61)	<0.001	1.40 (1.28–1.53)	<0.001
Others/Unknown	1640	0.78 (0.70–0.88)	<0.001	0.77 (0.70–0.84)	<0.001	1886	0.85 (0.73–0.99)	0.031	0.80 (0.71–0.91)	0.001
Differentiation										
Well	1993	Ref.		Ref.		1878	Ref.		Ref.	
Moderately	14,520	1.15 (1.02–1.30)	0.026	1.09 (1.00–1.18)	0.053	15,038	1.07 (0.90–1.28)	0.449	1.03 (0.90–1.18)	0.642
Poorly	3104	1.39 (1.21–1.59)	<0.001	1.25 (1.13–1.37)	<0.001	2700	1.48 (1.22–1.80)	<0.001	1.33 (1.15–1.55)	<0.001
Undifferentiated	235	1.32 (1.02–1.72)	0.038	1.33 (1.09–1.63)	0.005	472	1.64 (1.27–2.12)	<0.001	1.48 (1.19–1.82)	<0.001
LND										
<12	5221	Ref.		Ref.		1773	Ref.		Ref.	
≥12	14,631	0.67 (0.63–0.72)	<0.001	0.73 (0.69–0.76)	<0.001	18,315	0.54 (0.47–0.61)	<0.001	0.59 (0.53–0.65)	<0.001
AJCC T-stage										
T1	2451	Ref.		Ref.		2951	Ref.		Ref.	
T2	3647	1.30 (1.07–1.57)	0.007	1.15 (1.04–1.27)	0.007	3623	1.74 (1.30–2.34)	<0.001	1.45 (1.22–1.71)	<0.001
T3	11,319	3.13 (2.65–3.70)	<0.001	1.70 (1.56–1.86)	<0.001	10,319	4.47 (3.46–5.78)	<0.001	2.34 (2.02–2.71)	<0.001
T4	2435	6.50 (5.47–7.73)	<0.001	3.08 (2.79–3.41)	<0.001	3195	11.42 (8.72–14.76)	<0.001	4.97 (4.25–5.81)	<0.001
AJCC N-stage										
N0	12,018	Ref.		Ref.		12,110	Ref.		Ref.	
N1 ^†^	4804	2.22 (2.05–2.40)	<0.001	1.70 (1.60–1.80)	<0.001	5061	2.39 (2.14–2.67)	<0.001	2.07 (1.89–2.26)	<0.001
N2	3030	3.94 (3.62–4.28)	<0.001	2.67 (2.50–2.85)	<0.001	2917	4.60 (4.10–5.16)	<0.001	3.56 (3.24–3.92)	<0.001
CEA										
Negative	7488	Ref.		Ref.		7828	Ref.		Ref.	
Positive	3776	1.53 ( 1.41–1.65)	<0.001	1.53 (1.44–1.62)	<0.001	4007	1.69 (1.52–1.89)	<0.001	1.60 (1.46–1.74)	<0.001
Borderline	85	1.31 (0.86–1.97)	0.205	1.33 (0.98–1.81)	0.071	65	1.20 (0.62–2.32)	0.591	1.04 (0.59–1.83)	0.903
Others	8503	1.27 (1.19–1.37)	<0.001	1.32 (1.25–1.39)	<0.001	8188	1.40 (1.27–1.55)	<0.001	1.33 (1.23–1.44)	<0.001
Radiation therapy										
No	19,501	Ref.		Ref.		19,855	Ref.		Ref.	
Yes	351	1.34 (1.13–1.58)	0.001	1.29 (1.12–1.49)	0.001	233	1.30 (1.01–1.66)	0.041	-	-
Chemotherapy										
No/unknown	12,935	Ref.		Ref.		12,855	Ref.		Ref.	
Yes	6917	0.72 (0.67–0.77)	<0.001	0.64 (0.60–0.67)	<0.001	7233	0.54 (0.49–0.59)	<0.001	0.48 (0.44–0.52)	<0.001

Footnotes: SEER, the Surveillance, Epidemiology, and End Results; *LND*, lymph node dissection; *AJCC*, American Joint Committee on Cancer; *T*, primary tumor; *N*, regional lymph nodes; *M*, distant metastasis; *CEA*, carcinoembryonic antigen; *CSS*, cancer-specific survival; *OS*, overall survival; *HR*, hazard ratio; *CI*, confidence interval; ^†^ N1 included tumor deposits besides lymph node metastasis.

## Data Availability

The datasets supporting the conclusions of this article are available in the Surveillance, Epidemiology, and End Results (SEER) Program of the National Cancer Institute in the United States (http://www.seer.cancer.gov).

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
