# Peer review of "Poorer Survival in Patients with Cecum Cancer Compared with Sigmoid Colon Cancer"

_medicina, 2022, doi:10.3390/medicina59010045_

Round 1
Reviewer 1 Report
This manuscript has been well investigated in many cases.
The results were quite satisfying.
Author Response
Thank you for your positive comments. We really appreciate your suggestions and advices.
Reviewer 2 Report
I enjoyed reading this paper, however, I am surprised that caecal cancer prognosis is worse than sigmoid. Caecal cancer seems to demand less of meticulous surgical startegy (in comparison to ascending colon and hepatic flexurae tumors) but definitely standardization fo sigmoid cancer surgery is very straightforward.
Author Response

(The authors gave the same response as above.)

Reviewer 3 Report
Comments for authors:
The study entitled “Poorer survival in patients with cecum cancer compared with sigmoid
colon cancer” is novel and interesting. However, some minor concerns should be considered to improve the novelty as well as the flow of the manuscript:
1- Abstract: Acceptable
2- Keywords could be: Prognosis; Colon cancer; Cecum cancer; Sigmoid colon cancer, Compare
3- Introduction: Acceptable but could be better by wider literature review particularly related new & update articles from domestic researchers.
4- Methodology: Acceptable
5- Results: Acceptable
6- Discussion: Could be scientifically than present level using more and wider literature review and explanation of related and domestic survey and of course then using from them as Bibliography/References.
7- Bibliography/References: More domestic and update Ref(s). is necessary. Authors should do more literature review for using related domestic articles and therefore some more domestic Ref(s). (using from domestic Ref. is scientific discipline for publication of academic data.)
8- I could not see running title in manuscript. It is compulsory.
Final decision: Manuscript is acceptable for publication after minor correction
Important: Improve the quality your nice article by using new references like:
Sakjah S, Olsen ASF, Gundestrup AK, Born PW, Bols B, Ingeholm P, Kleif J, Bertelsen CA
Plane of mesocolic dissection as predictor of recurrence after complete mesocolic excision for sigmoid colon cancer: A cohort study.
Colorectal Dis. 2022
Nasseri Y, Wai C, Zhu R, Sutanto C, Kasheri E, Oka K, Cohen J, Barnajian M, Artinyan A
The impact of tumor location on long-term survival outcomes in patients with right-sided colon cancer.
Techniques in coloproctology. 2022
Zhao J, Zhu J, Huang C, Yuan R, Zhu Z
Impact of primary tumor resection on the survival of patients with unresectable colon cancer liver metastasis at different colonic subsites: a propensity score matching analysis.
Acta chirurgica Belgica. 2021
Surek A, Akarsu C, Gemici E, Ferahman S, Dural AC, Bozkurt MA, Donmez T, Karabulut M, Alis H
Risk factors affecting failure of colonoscopic detorsion for sigmoid colon volvulus: a single center experience.
International journal of colorectal disease. 2021
Wang L, Hirano Y, Ishii T, Kondo H, Hara K, Obara N, Yamaguchi S
Left colon as a novel high-risk factor for postoperative recurrence of stage II colon cancer.
World journal of surgical oncology. 2020
Sun Y, Mironova V, Chen Y, Lundh EPF, Zhang Q, Cai Y, Vasiliou V, Zhang Y, Garcia-Milian R, Khan SA, Johnson CH
Molecular Pathway Analysis Indicates a Distinct Metabolic Phenotype in Women With Right-Sided Colon Cancer.
Translational oncology. 2019
Nasseri Y, Wai C, Zhu R, Sutanto C, Kasheri E, Oka K, Cohen J, Barnajian M, Artinyan A
The impact of tumor location on long-term survival outcomes in patients with right-sided colon cancer.
Techniques in coloproctology. 2022
Huang J, Huang Q, Tang R, Chen G, Zhang Y, He R, Zu X, Fu K, Peng X, Xiao S
Hemicolectomy Does Not Provide Survival Benefit for Right-Sided Mucinous Colon Adenocarcinoma.
Frontiers in oncology. 2021
Tanabe M, Sawazaki S, Numata M, Koumori K, Maezawa Y, Aoyama T, Tamagawa H, Sato T, Yukawa N, Masuda M, Rino Y
[A Case of Sigmoid Colon Cancer Accompanied by Intestinal Malrotation Treated by Laparoscopic Surgery].
Gan to kagaku ryoho. Cancer & chemotherapy. 2019
Shaib WL, Zakka KM, Jiang R, Yan M, Alese OB, Akce M, Wu C, Behera M, El-Rayes BF
Survival outcome of adjuvant chemotherapy in deficient mismatch repair stage III colon cancer.
Cancer. 2020
Ben-Aharon I, Goshen-Lago T, Sternschuss M, Morgenstern S, Geva R, Beny A, Dror Y, Steiner M, Hubert A, Idelevich E, Shulman K, Mishaeli M, Man S, Liebermann N, Soussan-Gutman L, Brenner B
Sidedness Matters: Surrogate Biomarkers Prognosticate Colorectal Cancer upon Anatomic Location.
The oncologist. 2019
Falch C, Mueller S, Braun M, Gani C, Fend F, Koenigsrainer A, Kirschniak A
Oncological outcome of carcinomas in the rectosigmoid junction compared to the upper rectum or sigmoid colon - A retrospective cohort study.
Eur J Surg Oncol. 2019
Mochizuki K, Kudo SE, Ichimasa K, Kouyama Y, Matsudaira S, Takashina Y, Maeda Y, Ishigaki T, Nakamura H, Toyoshima N, Mori Y, Misawa M, Ogata N, Kudo T, Hayashi T, Wakamura K, Sawada N, Ishida F, Miyachi H
Left-sided location is a risk factor for lymph node metastasis of T1 colorectal cancer: a single-center retrospective study.
International journal of colorectal disease. 2020
Lyu LJ, Yao WW
Carcinoma located in a right-sided sigmoid colon: A case report.
World journal of clinical cases. 2022
Wang D, Agrawal R, Zou S, Haseeb MA, Gupta R
Anatomic location of colorectal cancer presents a new paradigm for its prognosis in African American patients.
PloS one. 2022
Author Response
Thank you for your positive comments. We really appreciate your suggestions and advices. In response to your comments, we have modified our manuscript. Detailed modifications are now listed hereinafter.
1、Keywords have been modified in our manuscript (sentences marked in red). Keywords: Prognosis; Colon cancer; Cecum cancer; Sigmoid colon cancer, Compare
2、Introduction has been modified in our manuscript (sentences marked in yellow).
Detailed modifications are now listed hereinafter.
Moreover, the prognostic differences between specific subsites (cecum, ascending co-lon, hepatic flexure, transverse colon, splenic flexure, descending colon, and sigmoid colon) have also been reported in a few studies [19-21].
3、Discussion have modified our manuscript (sentences marked in yellow).
Detailed modifications are now listed hereinafter.
And RCC were also showed a significantly worse 5-year survival (HR [95% CI]: 1.71 [1.10-2.64], P = 0.017) in a retrospective study of patients with colon cancer present in the Cancer Genome Atlas (TCGA) [14]. By analyzing the prognosis of patients with unresectable colon cancer liver metastasis, Zhao et al found that the risk of survival deterioration of RCC was significantly higher than that of LCC [26].
The present study showed that the survival difference between cecum cancers and sigmoid colon cancers is the largest among the six colon sites, which is like the results of Shaib et al [21]. By analyzing the data of patients with non-metastatic, inva-sive right-sided adenocarcinoma of the colon from 1988 to 2014 who underwent partial colectomy in SEER, Nasseri at al found that cecum cancers were prone to poorer dis-ease-specific survival (median 86.0, 93.0, and 89.0 months, respectively, p < 0.001) and OS (median not reached, p < 0.001) compared with cancers in other sites of the right colon[19]. In addition, Ben-Aharon et al also found the Oncotype Recurrence Score, a clinically validated predictor of recurrence risk in patients with stage II CRC, gradually decreased across the colon (cecum, highest score; sigmoid, lowest score; p = 0.04) [20].
It is worth noting that the prognostic difference between cecum cancer and sigmoid colon cancer is still significant after PSM.
Ward et al hypothesized three pathways through which tumor site impacts survival differences, including TNM stage, microsatellite instability (MSI), and other genetic drivers [14].
4、We have cited more new and domestic Ref(s) (sentences marked in yellow).
Detailed modifications are now listed hereinafter.
- Duraes, L.C.; Steele, S.R.; Valente, M.A.; Lavryk, O.A.; Connelly, T.M.; Kessler, H. Right colon, left colon, and rectal cancer have different oncologic and quality of life outcomes. Int J Colorectal Dis 2022, 37, 939-948, doi:10.1007/s00384-022-04121-x.
- Wang, L.; Hirano, Y.; Ishii, T.; Kondo, H.; Hara, K.; Obara, N.; Yamaguchi, S. Left colon as a novel high-risk factor for postoperative recurrence of stage II colon cancer. World J Surg Oncol 2020, 18, 54, doi:10.1186/s12957-020-01818-7.
- Ward, T.M.; Cauley, C.E.; Stafford, C.E.; Goldstone, R.N.; Bordeianou, L.G.; Kunitake, H.; Berger, D.L.; Ricciardi, R. Tumour genotypes account for survival differences in right- and left-sided colon cancers. Colorectal Dis 2022, 24, 601-610, doi:10.1111/codi.16060.
- Wang, D.; Agrawal, R.; Zou, S.; Haseeb, M.A.; Gupta, R. Anatomic location of colorectal cancer presents a new paradigm for its prognosis in African American patients. PLoS One 2022, 17, e0271629, doi:10.1371/journal.pone.0271629.
- Nasseri, Y.; Wai, C.; Zhu, R.; Sutanto, C.; Kasheri, E.; Oka, K.; Cohen, J.; Barnajian, M.; Artinyan, A. The impact of tumor location on long-term survival outcomes in patients with right-sided colon cancer. Tech Coloproctol 2022, 26, 127-133, doi:10.1007/s10151-021-02554-0.
- Ben-Aharon, I.; Goshen-Lago, T.; Sternschuss, M.; Morgenstern, S.; Geva, R.; Beny, A.; Dror, Y.; Steiner, M.; Hubert, A.; Idelevich, E.; et al. Sidedness Matters: Surrogate Biomarkers Prognosticate Colorectal Cancer upon Anatomic Location. Oncologist 2019, 24, e696-e701, doi:10.1634/theoncologist.2018-0351.
- Shaib, W.L.; Zakka, K.M.; Jiang, R.; Yan, M.; Alese, O.B.; Akce, M.; Wu, C.; Behera, M.; El-Rayes, B.F. Survival outcome of adjuvant chemotherapy in deficient mismatch repair stage III colon cancer. Cancer 2020, 126, 4136-4147, doi:10.1002/cncr.33049.
- Zhao, J.; Zhu, J.; Huang, C.; Yuan, R.; Zhu, Z. Impact of primary tumor resection on the survival of patients with unresectable colon cancer liver metastasis at different colonic subsites: a propensity score matching analysis. Acta Chir Belg 2021, 1-16, doi:10.1080/00015458.2021.1956799.
- Shao, Z.; Zheng, S.; Chen, C.; Lyu, J. Evaluation and Prediction Analysis of 3- and 5-Year Survival Rates of Patients with Cecal Adenocarcinoma Based on Period Analysis. International journal of general medicine 2021, 14, 7317-7327, doi:10.2147/ijgm.S334071.
5、We have modified our manuscript (sentences marked in green)
Running title: Poorer survival in cecum cancer
